# Temperate Zone Plant Natural Products—A Novel Resource for Activity against Tropical Parasitic Diseases

**DOI:** 10.3390/ph14030227

**Published:** 2021-03-07

**Authors:** Hamza Hameed, Elizabeth F. B. King, Katerina Doleckova, Barbara Bartholomew, Jackie Hollinshead, Haddijatou Mbye, Imran Ullah, Karen Walker, Maria Van Veelen, Somaia Saif Abou-Akkada, Robert J. Nash, Paul D. Horrocks, Helen P. Price

**Affiliations:** 1Centre for Applied Entomology and Parasitology, Keele University, Staffordshire ST5 5BG, UK; hamza83n@uomosul.edu.iq (H.H.); e.f.b.king@keele.ac.uk (E.F.B.K.); k.doleckova@centrum.cz (K.D.); haddijmbye@gmail.com (H.M.); iullah@hsph.harvard.edu (I.U.); w4r96@students.keele.ac.uk (M.V.V.); 2Department of Chemistry, College of Education for Pure Science, University of Mosul, Mosul, Iraq; 3Department of Biology, Faculty of Life Sciences, University of Hradec Králové, 500 03 Hradec Králové, Czech Republic; 4PhytoQuest Limited, Aberystwyth SY23 3EB, UK; barbara.bartholomew@phytoquest.co.uk (B.B.); jackie.hollinshead@yahoo.com (J.H.); robert.nash@phytoquest.co.uk (R.J.N.); 5MRC Unit The Gambia at LSHTM, Atlantic Boulevard, Fajara, Banjul PO Box 273, The Gambia; 6Harvard T.H. Chan School of Public Health, Harvard University, Boston, MA 02115, USA; 7School of Life Sciences, Keele University, Staffordshire ST5 5BG, UK; kwalker56789@gmail.com; 8Faculty of Veterinary Medicine, Alexandria University, Alexandria, Abis 10, Egypt; somaia_abuakkada@yahoo.com

**Keywords:** drug discovery, neglected tropical diseases, natural products, temperate zone, leishmaniasis, African trypanosomiasis, Surra, malaria

## Abstract

The use of plant-derived natural products for the treatment of tropical parasitic diseases often has ethnopharmacological origins. As such, plants grown in temperate regions remain largely untested for novel anti-parasitic activities. We describe here a screen of the PhytoQuest Phytopure library, a novel source comprising over 600 purified compounds from temperate zone plants, against in vitro culture systems for *Plasmodium falciparum*, *Leishmania mexicana*, *Trypanosoma evansi* and *T. brucei*. Initial screen revealed 6, 65, 15 and 18 compounds, respectively, that decreased each parasite’s growth by at least 50% at 1–2 µM concentration. These initial hits were validated in concentration-response assays against the parasite and the human HepG2 cell line, identifying hits with EC_50_ < 1 μM and a selectivity index of >10. Two sesquiterpene glycosides were identified against *P. falciparum*, four sterols against *L. mexicana*, and five compounds of various scaffolds against *T. brucei* and *T. evansi*. An *L. mexicana* resistant line was generated for the sterol 700022, which was found to have cross-resistance to the anti-leishmanial drug miltefosine as well as to the other leishmanicidal sterols. This study highlights the potential of a temperate plant secondary metabolites as a novel source of natural products against tropical parasitic diseases.

## 1. Introduction 

The Neglected Tropical Diseases (NTD) represent a diverse group of communicable diseases identified by the World Health Organisation (WHO) that cause significant morbidity and mortality amongst the poorest one billion people globally [1,2]. These diseases include viral, bacterial and parasitic diseases that share common challenges around the lack of investment in the development of new therapeutics to replace current treatments that may be affected by widespread resistance, poor tolerance due to toxicity and complicated, expensive and sometimes lengthy regimens. With future challenges around climate change along with an advocacy for a One Health approach integrating human and animal health with their environment, the importance of harnessing all our resources for drug development are pressing [3,4,5].

Natural products from plants have always been an important source of treatments for human disease—with records for the use of plant-based medicines found within the earliest records of humans to almost half the modern pharmacopeia being derived from natural products [6,7,8,9]. For tropical parasitic diseases, this impact is illustrated by the isolation of the active pharmaceutical quinine from the bark of the cinchona tree [10,11,12,13]. Whilst plant-based traditional medicine provides the basis for the success of many ethnopharmacological studies, a key limitation of this approach is that the search for treatments primarily focuses on plants indigenous to disease endemic regions.

The Phytopure library [14] represents a unique resource for the screening of anti-parasitic activity in the aetiological agents of human and animal tropical diseases. The library consists of compounds purified primarily from over 60 temperate zone plant families and represents a diverse range of plant secondary metabolite classes. Around two thirds of the compounds represent novel compounds not available in other commercial libraries. With some evidence for antiviral and antimicrobial action for compounds in the Phytopure library, this study set out to evaluate the potential for this novel resource as anti-parasitics. This study focused on three parasites from the Trypanosomatidae family (*Leishmania* and *Trypanosoma* species) representing our research interests in the search for new lead compounds to seed drug discovery efforts for the devastating human and livestock diseases they cause. In addition, the apicomplexan parasite *Plasmodium falciparum*, the aetiological agent of the most severe form of human malaria, was included to diversify the scope of the tropical parasitic diseases investigated.

Leishmaniasis is a spectrum of diseases caused by infection with the parasite *Leishmania* spp. which is transmitted by the bite of female sand flies. Leishmaniasis is endemic in 97 countries across the globe, with over 270,000 new cases reported to WHO in 2018 [15]. There are three main forms of the disease, depending largely on the species of parasite. The most severe form is visceral leishmaniasis (VL) or kala-azar, in which the parasites invade the liver and spleen; this condition is usually fatal unless treated. The most common form of the disease is cutaneous leishmaniasis (CL) which accounts for more than 75% of all cases. CL results in skin lesions which can take many months to heal, may be highly stigmatising and can leave permanent scarring [16]. A complication of CL is muco-cutaneous leishmaniasis (MCL) in which parasites invade and cause destruction of mucous membranes, particularly in the face and neck [16]. Treatment of all forms of leishmaniasis depends on the use of a small number of drugs which are largely expensive, toxic and difficult to administer. The pentavalent antimonials (e.g., meglumine antimonate) remain the first line treatment for CL despite high toxicity and resistant parasites emerging. The first line drug for VL in India is ambisome [17], a liposomal formulation of amphotericin B which shows reduced toxicity compared to the standard drug but is extremely expensive. Other options include miltefosine, pentamidine and paramomysin. Miltefosine is the only oral treatment for leishmaniasis but has teratogenic activity and so is contraindicated for pregnant women, and there have been major challenges in drug supply [18,19,20]. 

Human African Trypanosomiasis (HAT) is a vector-borne disease endemic to rural and impoverished areas of Sub-Saharan Africa. The causative agents of HAT are the subspecies of the parasite *Trypanosoma brucei*; specifically, *T. b. gambiense* and *T. b. rhodesiense* in West and East Africa, respectively. Control of the disease has been highly successful over the last 2 decades, with a decrease in new cases reported to WHO of 95% between 2000 and 2018 [21]. The decrease in cases will likely continue with the recent introduction of fexinidazole [22], the first orally available drug against *T. b. gambiense*. However, treatment of central nervous system (CNS) infection by *T. b. rhodesiense* still relies on melarsoprol, which is painful to administer and results in death of ~5% of patients from the treatment alone [23]. The subspecies *T. b. brucei* is non-infective to humans, is genetically tractable in the laboratory and has been used extensively as a model organism to study biogenesis and structure of organelles such as the flagellum and Golgi apparatus [24,25].

While there has been a large decrease in African trypanosomiasis in humans, infection due to African trypanosomes is still a considerable issue for livestock production. Surra is a major veterinary disease of livestock which is caused by the dyskinetoplastic parasite *Trypanosoma evansi*. Directly evolved from and closely related to *T. brucei*, *T. evansi* has lost the majority of its mitochondrial DNA and lacks the ability to carry out mitochondrial respiration [26]. The parasite is mechanically transmitted by biting insects, particularly tabanids and stomoxes [27]. This mechanism of transmission has enabled the parasite to spread beyond the African tsetse belt to the Middle East, North Africa, South East Asia and Latin America [28,29]. The parasite has a very broad host range and causes a wasting disease in camels, equines and dogs. While Surra is an economically devastating disease, it has not received as much attention as the human infective *Trypanosoma* species. The most common treatment, diminazene aceturate, has poor efficacy and tolerance in some animal species (e.g., horses) and has been used on livestock for decades, with resistant parasite strains now emerging. New effective therapies are urgently needed for this disease [27].

Human malaria falls outside of the NTD group, with significant efforts over recent years resulting in the introduction and continued development of new antimalarial drugs [30,31]. Increasing evidence of drug resistance against the frontline artemisinin combination therapies [32], however, highlights the demand for the continued search for new chemical classes to seed the drug discovery pipeline. 

This study set out to evaluate whether the PhytoQuest Phytopure library could act as a source of novel anti-parasitic leads. In doing so, the aim is to establish the potential of temperate zone plant natural products, a relatively uncharacterised source of leads for tropical parasitic disease research, as a novel resource in the fight against parasitic diseases.

## 2. Results and Discussion

### 2.1. Screening the PhytoQuest Phytopure Library for Growth Inhibition Identifies Multiple Hits across Different Parasites

634 compounds from the PhytoQuest Phytopure temperate plant natural product library were screened for growth inhibitory activity against *L. mexicana* axenic amastigotes, *T. brucei* bloodstream form, intraerythrocytic asexual *P. falciparum* (at 2 μM) and *T. evansi* bloodstream form parasites (at 1 μM). Figure 1 summarizes the results across all parasites using a heatmap where the most potent compounds are represented in dark grey/black and the least potent in light grey (see Appendix A for raw data). Our initial hits were defined as reducing growth of the parasite by ≥50%. 

For *L. mexicana* this threshold was reduced to inhibiting growth by ≥80% survival due to the large number of hits (64 in total) using the ≥50% criteria. A total of 14 compounds were therefore taken forward as initial hits against *L. mexicana*, including four sterols (700022, 700107, 700136, 700240), seven sesquiterpenes (700756, 701154, 701155, 701157, 701158, 701159, and 701212) and three aromatic diynes (701044, 701241, and 701248). Of these 14 compounds, seven were identified as an initial hit in at least one of the other three parasite species in this screen. 

There were 18 initial hits against *T. brucei*, including three flavonoids (700585, 700586, 701082), four bisobolane sesquiterpene glycosides (700035, 700042, 700046, 700048); six additional sesquiterpenes (701154, 701155, 701156, 701157, 701158, 701159), an abietane diterpenoid (700014), an aromatic diyne (701241), an iridoid monoterpene (701145), a coumarin (700867) and a glycoside (700004). Of these 18 hits, nine were active against at least one other parasite species in the screen. 

The *T. evansi* screen yielded 15 initial hits, including three sesquiterpenes (701155, 701157, 701158), two bisobolane sesquiterpene glycosides (700042, 700046), two abietane diterpenoids (700063, 700454), a terpenoid glycoside (700458); two flavonoids (700326, 701088), a sterol (700016), a glycoside (700004) and a lignan (700144). Of the 15 hits, six were identified as an initial hit in at least one of the other three parasite species in this screen. Of particular note, despite the close phylogenetic relationship of the two *Trypanosoma* genus parasites in the screen, only six compounds were identified as initial hits in both *T. brucei* and *T. evansi*. These were five sesquiterpenes (701155, 701157, 701158), including two bisobolane sesquiterpene glycosides (700042, 700046), as well as the glycoside (700004). 

The *P. falciparum* screen identified eight initial hits, six of which were sesquiterpenes (700,535, 701158) with four of these being bisobolane sesquiterpene glycosides (700042, 700046, 700048, 700104). Of the eight hits, five were identified as an initial hit in at least one of the other three parasite species in this screen. 

The total number of compounds identified as having growth inhibitory activity in at least one parasite species was 38/634, equating to 6% of the PhytoQuest Phytopure library (Figure 2, Appendix A). Of these 38 compounds, 71% were terpenes with all four parasites showing sensitivity to multiple terpenoids. This is unsurprising, as terpenoids, and sesquiterpenes in particular, have previously been reported to have broad anti-plasmodial and anti-kinetoplastid activity [33,34]. For example, the sesquiterpene 701158, isolated from *Arnica montana* (commonly known as Mountain Tobacco) was identified as a hit in all parasite screens. Similarly, bisobolane sesquiterpene glycosides isolated from *Phyllanthus acuminatus,* and similar in structure to the phyllanthostatins [35,36], were identified as initial hits against all parasites in this study except *L. mexicana*. The abietane diterpenoids 700014 and 700063 were identified in the *T. brucei* and *T. evansi* screens, respectively, and represent a class of plant-derived natural product known to have a wide range of growth inhibitory activity against *L. donovani*, *L. major*, and *P. falciparum* [34,37,38]. Whilst neither 700014 or 700063 showed activity against *P. falciparum*, they did decrease *L. mexicana* survival by some 70% and thus fell just below the selection threshold. 

The rearranged abietane diterpenoid 700454, which was identified in the *T. evansi* screen, is constitutionally identical to the previously identified leriifoliol [39]. Stereochemistry has not been assigned in this study, and thus it is unknown whether they are identical stereochemically. Interestingly, leriifoliol has previously been isolated and tested against protozoan parasites, exhibiting micromolar activity against *T. b. rhodesiense* and submicromolar activity against *P. falciparum* [39]. However, in our screen 700454 only reduced *P. falciparum* growth by 5%. Understanding whether this could be due to 700454 having divergent stereochemistry to leriifoliol that affects the growth inhibitory activity against *P. falciparum*, or whether the difference occurs as a result of the use of the multidrug-resistant Dd2^Luc^ strain in the screen, compared to the drug sensitive NF54 strain in the original report, could be useful in terms of understanding any target.

Four of the lanosterone-like sterols (700022, 700107, 700136, 700240) only inhibited growth in *L. mexicana*. Similar sterols, such as pistagremic acid, have also shown leishmanicidal activity [34,40] and have also been implicated in lipase and anticancer activity [41,42]. Interestingly, 700016, another lanosterone-like sterol, was effective against *T. evansi* and decreased *L. mexicana* survival by more than 50% (Appendix A). The predominant structural difference between the leishmanicidal and trypanocidal sterols is that the lactone headgroup, a 3-methyl-2(5H)-furanone in 700016 is replaced instead with a 5-hydroxy-3-methyl-2(5H)-furanone in 700022, 700107 and 700136 and a ring-opened lactone in 700240 (Figure 2, Appendix A). The absence of the 5-hydroxy group appears to increase the activity of these sterols in *T. evansi,* relative to a decrease their activity in *L. mexicana*. *Abies procera* derived lanosterone-like sterols from the PhytoQuest Phytopure library have previously been identified as potent inhibitors of the helminth parasites *Schistosoma mansoni* and *Fasciola hepatica* [43], highlighting this class of molecule as a potential lead for multiple parasitic diseases. 

### 2.2. Demonstrating Selectivity of These Anti-Parasitic Hits

Initial hits from the parasite panel screen were taken forward in log concentration versus normalised response assays to estimate the half maximal effective concentration (EC_50_) in hit species as well as the half maximal cytotoxic concentrations (CC_50_) in the human hepatoma cell line HepG2 as a preliminary test for selectivity. Using CC_50_/EC_50_ to define the selectivity index (SI), the selectivity for each compound in a defined species compared to a human cell line was estimated and compared to the EC_50_ (Figure 3). From this analysis, the application of new thresholds (EC_50_ < 1 μM and an SI of ≥10) was used to refine the hits for each parasite species (Appendix A). 

The four lanosterone-like sterols derived from the Noble Fir *Abies procera* (700022, 700107, 700136) and the Grand Fir *Abies grandis* (700240) were the only initial compounds taken forward for *L. mexicana* due to their nanomolar activity and apparent low cytotoxicity (Figure 3a). Of these sterols, 700240 was the least potent. Understanding the impact of the ring-opened lactone head group in this compound compared to the other three warrants further analysis in exploring their structure-activity relationships. These four sterols were also tested for growth inhibitory activity against *L. donovani* axenic amastigotes. The similar potency of their growth inhibitory activity (Figure 4a) suggests they may have a broader leishmanicidal activity beyond that reported here for *L. mexicana*. As *Leishmania* spp. amastigotes typically reside with macrophages, further cytotoxicity assays with these sterols were performed against an activated monocyte cell line (THP-1) (Figure 4b). The sterols generally had CC_50_ > 20 μM providing SI of >20, although they were more cytotoxic against THP-1 than HepG2 cell lines (Appendix A). Following the observations made above for the ring-opened lactone head group in 700240, this compound also had the lowest potency of these four sterols in *L. donovani* and the lowest selectivity (SI of 21) against THP-1 (CC_50_ of 13.5 μM) cell lines. 

Five compounds were identified with acceptable selectivity and submicromolar activity in *T. brucei* (Figure 3b). These were three bisobolane sesquiterpene glycosides (700035, 700046 and 700048), the iridoid monoterpene (701145) and the flavonoid (700585). Further analysis of 700046, 700048 and 700585 was not conducted as, whilst the preliminary selectivity against HepG2 was >10, the actual CC_50_ was <10 μM and suggested a general toxicity risk (Appendix A). Compound 701145 is an iridoid monoterpene with a deaminated tyrosine (Figure 2) that was isolated from Bogbean, *Methyanthes trifoliata*, with an EC_50_ and CC_50_ of 0.52 μM and 27.8 μM, respectively. Compound 700035 was the least cytotoxic of the bisobolane sesquiterpene glycosides (HepG2 CC_50_ of 15.2 μM) although it is apparently quite potent with an EC_50_ of 0.35 μM. The 700035 was isolated from the Jamaican Gooseberry tree *Phyllanthus acuminatus* and is very similar to the previously synthesised (+)-Phyllanthocin 3 [35]. The structural variation is an acetate group on the second sugar being on C4 in 700385 rather than on C3 in (+)-Phyllanthocin 3. Of the bisobolane glycosides identified in this screen, 700035 is the only one with a ring-opened epoxide, suggesting that this motif warrants further investigation, particularly in *T. evansi* and *P. falciparum*. 

Five of the *T. evansi* initial hits were indicated for further analysis, although the bisobolane sesquiterpene glycoside 700046 was discounted due to the demonstrated intrinsic HepG2 antiproliferative activity (Figure 3c, Appendix A). Both 700144 and 700513 had low or no detectable growth inhibitory activity in HepG2 up to 100 μM. Compound 700144 is a lignan isolated from *Hewittia sublobata* (Figure 2), with some similarities in structure to other lignans such as arctigenin and matairesinol, which are known to have broad antiproliferative activity [44,45,46]. The activity of 700144 is likely different to that of arctigenin, however, as the latter exhibits a potent antiproliferative activity against HepG2 cells [47]. Lignans similar to 700144 have been shown to have in vitro activity against *L. donovani* axenic amastigotes, *T. b. rhodesiense* and *P. falciparum*, though not at the levels of potency observed here against *T. evansi* [48]. Compound 700513, a labdane-like diterpenoid compound isolated from the tropical lilac *Cornutia grandiflora* (Figure 2, Appendix A), displayed similarly promising potency and selectivity towards *T. evansi*. While not as selective as 700144 and 700513, the flavonoid glycoside 701088 (Figure 2) also exhibited high levels of growth inhibition in *T. evansi* and selectivity against HepG2 (Figure 3c). This class of compound has previously been reported to have both antiviral and antimalarial activity, as well as having an important role as a secondary metabolite in the oak tree *Quercus ilex* [49,50,51]. The glycoside 700004 (Figure 2) satisfied the thresholds of potency and selectivity, adding to the structural diversity of the *T. evansi* potential leads identified in this study (Figure 3c).

The three compounds that demonstrated potency and selectivity for *P. falciparum* were bisobolane sesquiterpene glycosides (700046, 700048, 700104) (Figure 3d). The potency of these compounds was also tested against the more sensitive *P. falciparum* 3D7 strain, and found to have similar sub-micromolar activity with 700046 being the most potent in both strains (Figure 5a). To further explore the pharmacodynamics of their activity the initial rate of kill was determined using the bioluminescence relative rate of kill (BRRoK) assay [52,53] and compared to the rates of kill for atovaquone (ATQ, slow killing compound with 48 h lag in action) and dihydroartemisinin (DHA, rapid initial rate of kill with no lag in action) (Figure 5b, Appendix A) [52]. All three bisobolane sesquiterpene glycosides had a similar initial rate of kill (Figure 5c, Appendix A). This is not surprising as the rate of kill is a result of the mechanism of cell death and, given their structural similarity, these three bisobolane sesquiterpene glycosides likely share the same target(s), albeit with some differences in affinity. The rate of kill, relatively, falls between that of atovaquone and dihydroartemisinin and is more similar to that of chloroquine [52,53]. Unfortunately, all three compounds showed intrinsic toxicity in our preliminary selectivity screen against HepG2 (CC_50_ of approximately 2 μM, Appendix A) and were not developed as antiparasitic lead compounds any further. 

Whilst sesquiterpenes generally showed submicromolar activity against all the trypanosomatids screened here, they demonstrated a similar potency against the HepG2 cell line and had a preliminary SI of <10 in all cases. For example, compound 701158, which was identified as a hit against all parasites in the initial screen, had a HepG2 CC_50_ of 0.8 μM, compared to an estimated 1 μM activity in *P. falciparum*. The only sesquiterpene that had a CC_50_ ≥ 10 μM in the HepG2 cell line was the bisobolane glycoside 700035. An important lesson from this screen was that a pan-parasite panel activity was almost always associated with toxicity in our preliminary cytotoxicity screens against HepG2. 

Overall, this screen has identified potential lead compounds in *L. mexicana*, *L. donovani*, *T. brucei* and *T. evansi* which warrant further investigation. No compounds of interest, however, were identified against *P. falciparum*, despite the relatively low stringency of our potency and selectivity thresholds. 

### 2.3. L. mexicana Parasites with Resistance to 700022 Have Cross-Resistance to Miltefosine but Not to Amphotericin B

A 700022-resistant line of *L. mexicana* (r22) was generated using a previously reported method [54] (Appendix A). Briefly, promastigote parasites were cultured with 700022 over 28 weeks, with incremental increases in compound concentration as the EC_50_ in promastigotes increased from 11.5 μM to 85.6 μM in promastigotes (Figure 6a) and from 0.24 μM to 10.1 μM in axenic amastigotes (Figure 6b). All leishmanicidal sterols identified were assayed against r22 after 8 weeks of 700022 pressure and cross-resistance was found against all four compounds, each with a similar resistance index of between 4–5 (Figure 6c). Following 28 weeks of selection, the morphology of the r22 promastigotes was compared to that of WT (Figure 6d). The r22 strain appear to have a significant five-fold reduction in the mean length of the flagellum (Figure 6e), suggesting either a fast growth phenotype or dysfunction in flagellar development [55]. The body area of both promastigotes and amastigotes was also assessed, noting a small (c 20%) decrease in the body area of r22 parasites compared to WT strains (Appendix A). While further investigation into the virulence and transmissibility of this 700022-resistant line was not performed, shortened flagella do not necessarily prevent colonisation of the sandfly vector and thus transmissibility might not be compromised [55,56]. 

To assess the effect of resistance to 700022 in the r22 line compared to current therapies for leishmaniasis, the EC_50_ for amphotericin B, miltefosine and pentamidine were assessed in WT and r22 lines (Figure 6f). Whilst there was no apparent change in EC_50_ for pentamidine and amphotericin B (resistance indices of 0.75 and 1.22, respectively), there was a significant shift in the miltefosine resistance index (17.9). A further issue arose from preliminary *L. mexicana* intramacrophage assays also suggesting that these four sterols were not as potent at killing the amastigote parasites within the activated monocyte THP-1 cell line (Appendix A). Evaluation of the EC_50_ for 700022 in a *L. mexicana* NanoLuc PEST strain [57] reveals that the EC_50_ in intramacrophage amastigotes is 10.7 µM, reducing the selectivity of this compound over THP-1 and HepG2 to factors of 1.6 and 2.7, respectively. Whilst the miltefosine cross-resistance and reduction in potency against intramacrophage amastigotes indicates that these sterols are not appropriate for further development, our findings emphasise the importance of screening hits for cross-resistance against current therapies at an early stage. 

## 3. Materials and Methods 

### 3.1. Culture of Parasites and Human Cell Lines

*L. mexicana* strain MNYC/BZ/62/M379 and *L. donovani* strain LdBOB (a clonal line from strain MHOM/SD/62/1S-CL2) were maintained in vitro in the procyclic promastigote stage. Parasites were cultivated at 26 °C in Schneider’s medium (Gibco) pH 7.0 containing 10% FBS, 100 U/mL penicillin (Lonza) and 100 μg/mL streptomycin (Lonza) as previously described [57]. Differentiation into axenic amastigotes was performed as described previously [58]. Briefly, axenic amastigotes were cultivated at 32 °C in Schneider’s medium pH 5.5 containing 10% FBS, 100 U/mL penicillin (Lonza) and 100 μg/mL streptomycin (Lonza). Bloodstream form *T. brucei brucei* strain Lister 427 and *T. evansi* strain Antat 3/3 [59] were maintained in vitro at 37 °C with 5% (*v*/*v*) CO_2_ in HMI-11 medium supplemented with 10% FBS, as described elsewhere [60,61]. Intraerythrocytic cultures of *P. falciparum* Dd2^Luc^ strain [62] were maintained at 37 °C in a 1% O_2_:3% CO_2_:96% N_2_ atmosphere in RPMI-1640 containing 37.5 mM HEPES, 5 mM NaOH, 10 mM D-glucose, 2 mM L-glutamine, 100 μM hypoxanthine, 25 mg/mL gentamicin sulfate, 5% human serum and 5% albumax-II and 2% haematocrit erythrocytes [63,64]. When required, cultures were synchronised to ring stages using the sorbitol lysis method [65]. Preliminary cytotoxicity screening was assessed using HepG2 cells. These cells were maintained in vitro at 37 °C with 5% (*v*/*v*) CO_2_ in DMEM (Sigma) pH 7 supplemented with 10% FBS, 100 U/mL penicillin (Lonza) and 100 μg/mL streptomycin (Lonza), as previously described [66,67]. The human monocyte cell line THP-1 [68] was maintained in vitro by culturing at 37 °C with 5% (*v*/*v*) CO_2_ in complete RPMI medium (Dutch modified RPMI-1640 (Gibco) containing 10% FBS and 2 mM L-glutamine (Gibco)). Differentiation of THP-1 cells into macrophages was performed by seeding 2.5 × 10^5^ cells/mL in complete RPMI media, supplemented with 20 ng/mL phorbol 12-myristate 13-acetate (PMA), followed by incubation at 37 °C with 5% (*v*/*v*) CO_2_ for 24 h [69]. 

### 3.2. Screening

The PhytoQuest Phytopure library is a commercially available collection of purified compounds isolated predominantly from temperate zone plants, with structures confirmed by NMR and mass spectrometry. The library was provided as 634 non-polar compounds (1 mg/mL in DMSO). Initial screening was performed in a 96-well plate with 200 μL of parasite culture. The initial screen used each compound at a final concentration of 2 μM for *L. mexicana* axenic amastigotes, *T. brucei* and *P. falciparum*, and at 1 μM for *T. evansi*. *L. mexicana* axenic amastigotes were plated at 1 × 10^6^ cells/mL, incubated for 72 h, and the Alamar Blue assay was used to assess parasite growth [70]. Each compound was tested in triplicate, with two biological replicates (*n* = 6). *T. brucei* and *T. evansi* were plated at 1×10^5^ cells/mL for 48 h, and the Alamar Blue assay was used to assess parasite growth [70]. Each compound was tested in triplicate with two biological replicates (*n* = 6). Intraerythrocytic *P. falciparum* in the trophozoite stage were seeded at 0.5% parasitaemia and 2% haematocrit for 48 h, and the Malaria SYBR-Green I fluorescence assay was used to assess parasite growth [71]. Each compound was tested in duplicate with two biological replicates (*n* = 4). Negative controls comprised an equivalent volume of DMSO (equivalent of 1% *v/v*) to normalise the growth data. Data points >100% growth and <0% growth were tabulated as 100% and 0%, respectively. Some compounds were not tested in all parasite lines as there was insufficient material remaining. 

### 3.3. Log Concentration v Normalised Response Curves to Estimate EC_50_

EC_50_ values were determined by serial two-fold dilution of each compound in 96 well plates. Culture conditions and viability assays were as stated for the initial screen, with some exceptions. *L. mexicana* and *L. donovani* axenic amastigotes were seeded at 2 × 10^6^ cell/s mL, *T. brucei* and *T. evansi* were seeded at 2 × 10^5^ cells/mL. For cytotoxicity studies, HepG2 cells were seeded at 1 × 10^5^ cells/mL, incubated for 48 h and viability was assessed using the Alamar Blue assay [66,67]. Differentiated THP-1 cells were seeded at 5 × 10^4^ cells/mL, incubated for 48 h, and viability was assessed by the Alamar Blue assay [57]. Normalised growth and viability data (compared to untreated controls) were plotted against Log10 concentration to estimate EC_50_ in GraphPad (Prism v6). Bioluminescence relative rate of kill assays in asexual intraerythrocytic *P. falciparum* Dd2^Luc^ were carried out as described [52,53,64]. 

### 3.4. Generation of 700022-Resistant L. mexicana 

*L. mexicana* parasites were grown in increasing concentrations of 700022 in a stepwise manner, as previously described [54]. Briefly, promastigotes were cultivated in a starting concentration of 11.5 μM of 700022 (the EC_50_ of this compound against WT *L. mexicana* procyclic promastigotes) and passaged at this concentration of inhibitor until the growth rate matched that of the WT *L. mexicana*. A dose response assay of the newly selected promastigotes was completed as stated above, with the exception of the seeding density of 1 × 10^5^ cells/mL. At this point, the drug pressure was increased to the concentration of the new EC_50_ value. This process was completed over 28 weeks until the EC_50_ reached 85.5 μM. 

### 3.5. Morphological Analysis

Scanning electron microscopy (SEM) was performed as previously reported. Briefly, WT and r22 *L. mexicana* promastigote parasites were washed 3 times in serum-free Schneiders media (pH 7.0) and once with PBS. Samples were seeded onto poly-L-lysine coated 12 mm coverslips, washed once with PBS and then fixed with 2.5% glutaraldehyde in 0.1 M sodium cacodylate with 2 mM calcium chloride (pH 7.4) for 2 h, then processed and imaged as described previously [72]. 

Cell volume and flagella length was calculated using immunofluorescence microscopy. Wild-type and r22 *L. mexicana* promastigote and axenic amastigote parasites were fixed in 4% (*w*/*v*) paraformaldehyde (PFA), adhered to poly-L-lysine slides, permeabilised in 0.1% Triton X-100 in PBS, blocked with Image iT FX Signal Enhancer (Life Technologies). Cells were probed with anti-α-Tubulin diluted 1:250 in PBS followed by anti-mouse Alexa Fluor 488 diluted 1:200 in PBS. DNA was stained using 10 μg/mL DAPI, then washed and mounted. Slides were analysed using the EVOS FL cell imaging system (ThermoFisher Scientific). Flagellum length and body area of >200 randomly selected parasites (WT and r22) were measured using ImageJ (version 1.48). Statistical analysis was completed in R; statistical difference was assessed using the Mann–Whitney U test in the psych package [73,74].

## 4. Conclusions 

A screen of the PhytoQuest Phytopure temperate plant natural product library revealed a number of promising hit compounds against the kinetoplastid parasites screened (Figure 7). Further analysis of a series of leishmanicidal sterols, including the generation of a resistant parasite line, revealed cross-resistance with miltefosine, an important frontline therapy with ongoing reports on the evolution of resistance [19]. As a number of sterols are being explored for their potential as possible leads against *Leishmania* spp. [34], this study highlights the importance of including screening novel compounds against drug-resistant lines early in the hit-to-lead process. 

While *T. brucei* drug discovery for human use has been relatively well developed for many years, development of therapies against veterinary pathogens such as *T. evansi* have been neglected. The lack of overlap between hits with potency against the two *Trypanosoma* species used here suggests that drugs developed for HAT may not necessarily readily translate as a treatment for Surra. Much more fundamental work in the screening of novel candidates that target *T. evansi* is needed, as well as further detailed investigation of selectivity using a panel of human and mammalian cell lines. This study has contributed to this imperative, identifying 700513 and 700144 as highly selective and potent hits against *T. evansi* which warrant further investigation. 

## Figures and Tables

**Figure 1 pharmaceuticals-14-00227-f001:**
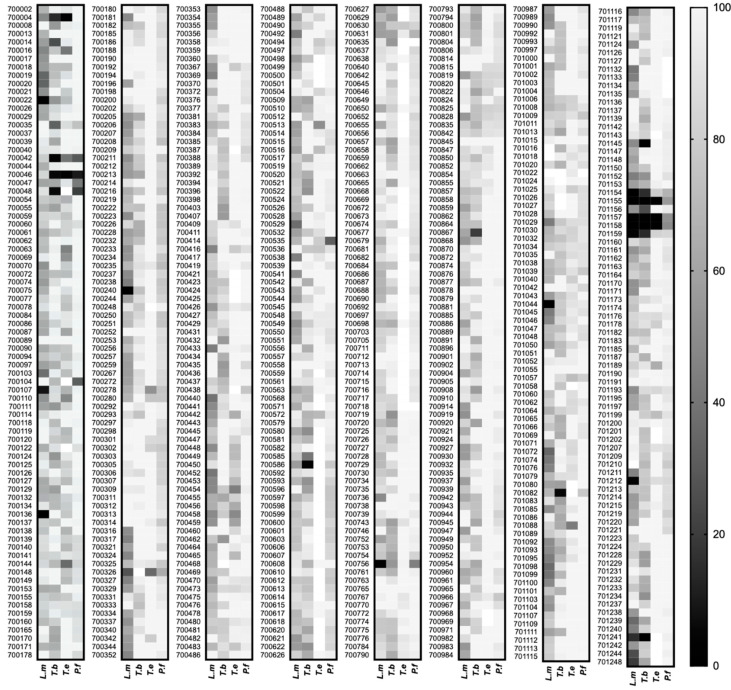
Heatmap of PhytoQuest Phytopure temperate natural product screen against protozoan parasites. In total, 634 compounds were screened against *L. mexicana* (*L.m*) axenic amastigotes, *T. brucei* (*T.b*) bloodstream form and *P. falciparum* (*P.f*) intraerythrocytic trophozoites at 2 μM and *T. evansi* (*T.e*) bloodstream form parasites at 1 μM. Survival (reported as % of DMSO control) of each parasite cell line is represented as a spectrum from black to light grey (see scale to right), with the black cells reporting hits taken forward in this study. Blank cells indicate where no data were collected for that compound.

**Figure 2 pharmaceuticals-14-00227-f002:**
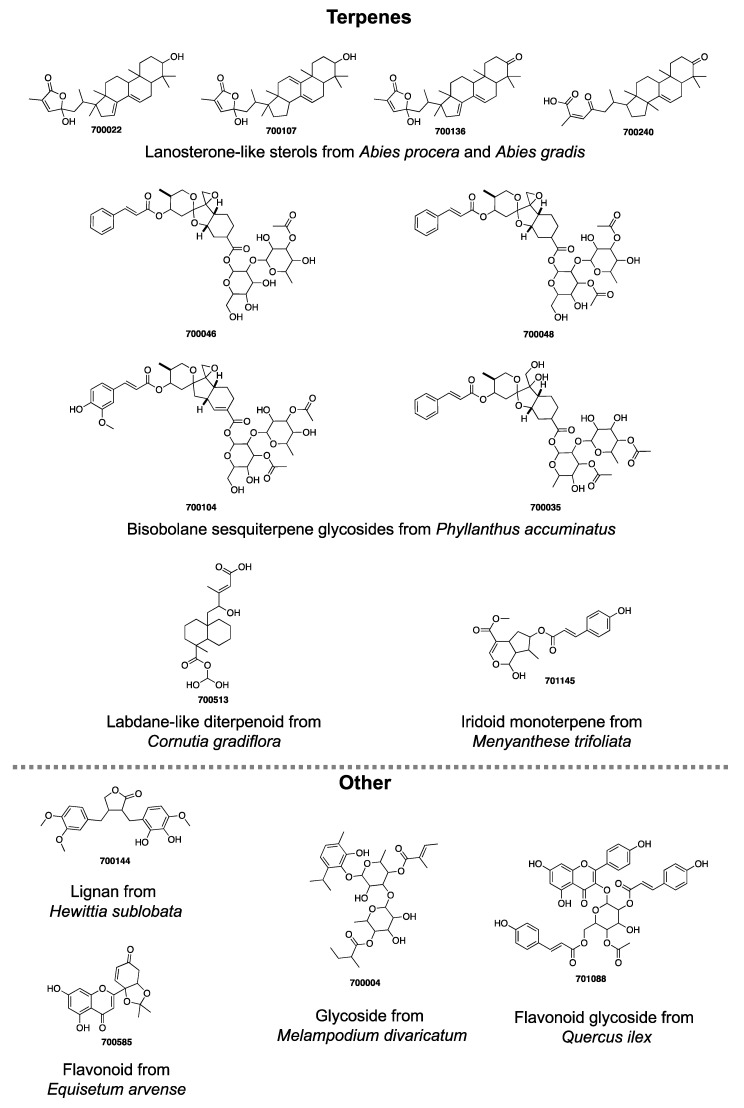
Structures of compounds which had <1 μM activity against the relevant parasite cell line, with an SI of ≥10.

**Figure 3 pharmaceuticals-14-00227-f003:**
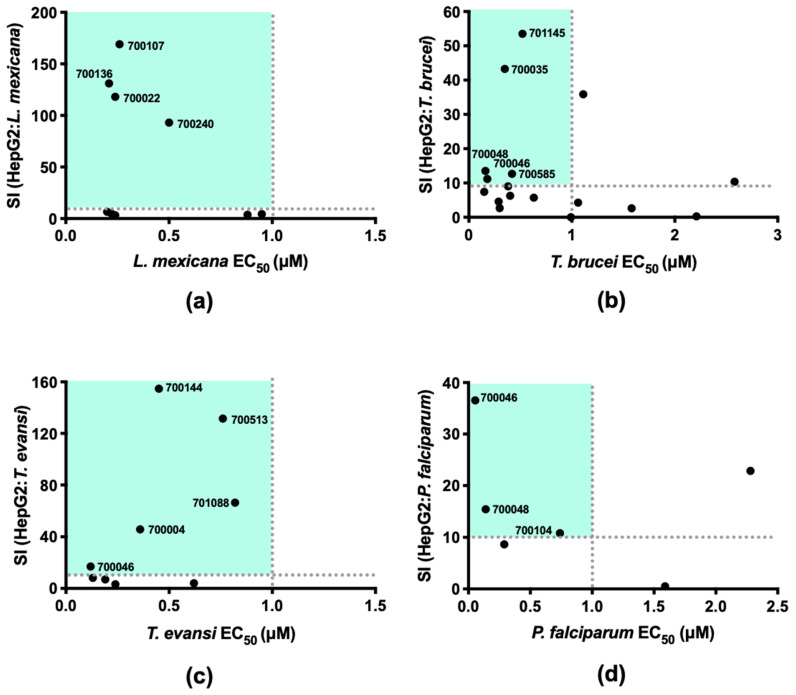
Comparison of potency and cytotoxicity of hit compounds. Comparison of the EC_50_ to the selectivity index (SI) of hit compounds in (**a**) *L. mexicana*, (**b**) *T. brucei*, (**c**) *T. evansi* and (**d**) *P. falciparum.* The SI reports the CC_50_ of the compound in HepG2 cell line divided by the EC_50_ of the reported parasite line. The grey dotted lines indicate the preferred potency and SI (≥10) thresholds, with hits taken forward from the top left quadrant only. The green box highlights the potential lead compounds that fall within these thresholds.

**Figure 4 pharmaceuticals-14-00227-f004:**
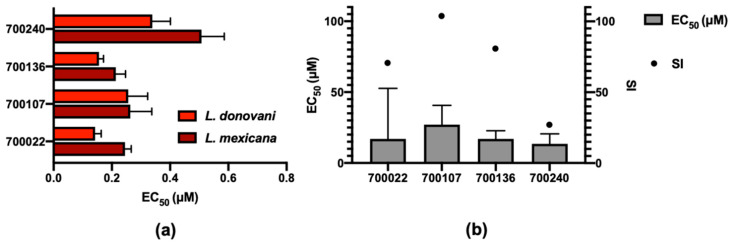
Sterol activity against two *Leishmania* species and an activated monocyte cell line. (**a**) Mean EC_50_ (μM) of the reported sterol compounds against *L. donovani* and *L. mexicana* axenic amastigotes. Error bars represent upper limits of the 95% confidence interval. (**b**) EC_50_ values (left axis) and selectivity index (SI, right axis) of the same sterols against an activated monocyte cell line (THP-1). Error bars represent upper limits of the 95% confidence interval for the EC_50_ and the SI represents the selectivity of these compounds against *L. mexicana* compared to THP-1.

**Figure 5 pharmaceuticals-14-00227-f005:**
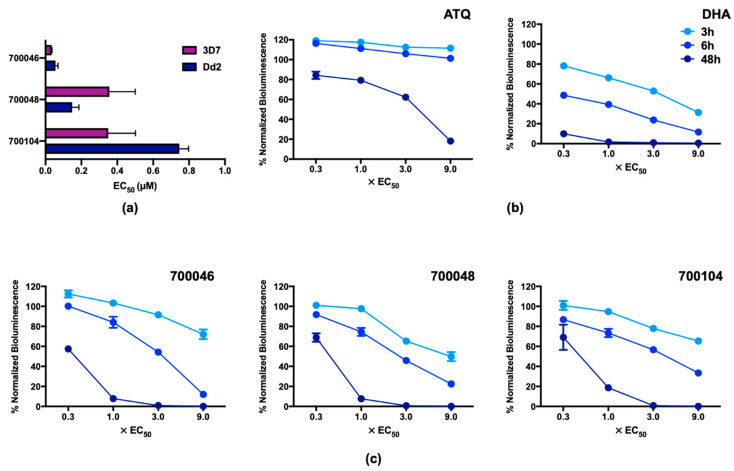
Potency and rate of kill of bisobolane sesquiterpene glycoside hits against *Plasmodium falciparum*. (**a**) Comparison of EC_50_ values of bisobolane sesquiterpene glycoside hits against intraerythrocytic *P. falciparum* Dd2^Luc^ and 3D7 cell lines. Time course of rate of kill for (**b**) control antimalarial compounds atovaquone (ATQ) representing a slow rate of kill and dihydroartemisinin (DHA) representing a fast rate of kill. against (**c**) the bisobolane sesquiterpene glycoside hits. Time course data (3, 6 and 48 h) show the normalized bioluminescence signal (compared to an untreated control at the same timepoint) following exposure to a fold-EC_50_ exposure of the indicated compound. Error bars represent upper and lower limits of the StDev (*n* = 9). See Appendix A for concentrations used.

**Figure 6 pharmaceuticals-14-00227-f006:**
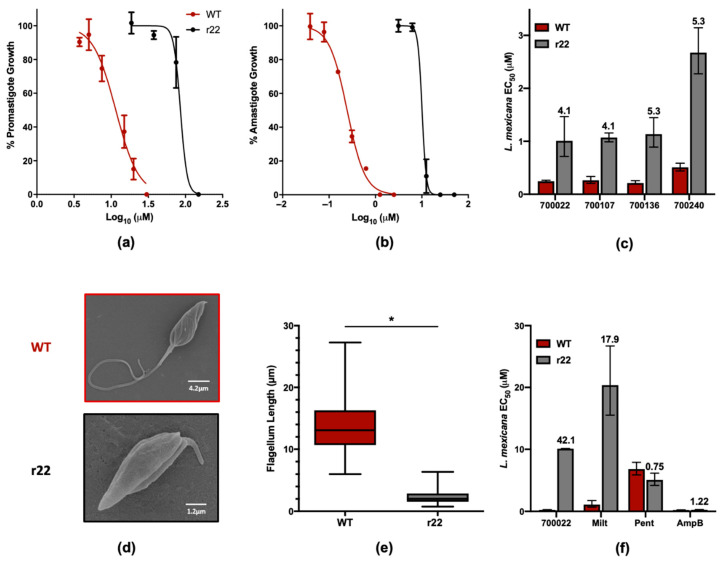
Comparison between wildtype and 700022-resistant (r22) *L. mexicana* cell lines. Log concentration versus normalised (compared to untreated control) growth curves for wildtype (WT) and corresponding 700022-selected (r22) *L. mexicana* (**a**) promastigotes and (**b**) axenic amastigotes. Error bars represent upper and lower limits of the StDev (*n* = 9). (**c**) Average EC_50_ of WT and r22 after 8 weeks of 700022 selection pressure against 700022 and structurally related sterol hits against *L. mexicana*. The resistance index (EC_50_ in r22/EC_50_ in WT) are shown for each compound. (**d**) Scanning electron micrographs of WT and r22 promastigotes illustrating the shortened flagellum in r22 parasites. (**e**) Analysis of flagellum length in WT and r22 promastigotes (*n* = 287 of each strain) reported using a box and whisker plot (boxes represents the 25–75th percent distribution with the mean as a horizontal line, whiskers represent the distribution of all values) with a statistically significant difference (Mann–Whitney U test, *p*-value < 2.2 × 10^−16^). (**f**) Mean EC_50_ of a panel of indicated compounds against WT and r22 after 28 weeks of 700022-selection pressure. Error bars represent upper and lower limits of the StDev (*n* = 9). Miltefosine, Milt (concentration range used against WT and r22 was 0.16–20 μM and 1.6–50 μM, respectively); Pentamidine, Pent (concentration range used against both WT and r22 was 0.78–50 μM); Amphotericin B, Amp B (concentration range used against both WT and r22 was 0.078–1.3 μM). The resistance index (EC_50_ in r22/EC_50_ in WT) are shown for each compound.

**Figure 7 pharmaceuticals-14-00227-f007:**
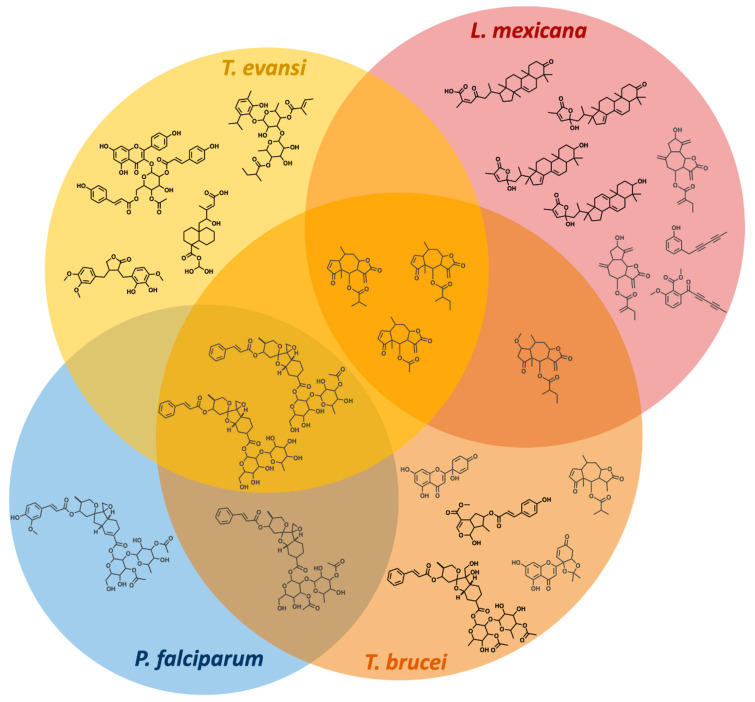
Comparison of the PhytoQuest Phytopure library hits across the four parasite species investigated. All compounds shown have an EC_50_ ≤ 1 μM in the respective species reported here. Compounds reported in black indicates a high selectivity (SI > 20) and/or a low cytotoxicity in HepG2 (CC_50_ > 20 μM). Compounds in grey indicates a low selectivity (SI ≤ 20) and/or toxicity in HepG2 (CC_50_ ≤ 20 μM).

## Data Availability

Data is contained within the article or Appendix A. The full screening dataset for this study is available in Appendix A as listed above.

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
