# Peer review of "Temperate Zone Plant Natural Products—A Novel Resource for Activity against Tropical Parasitic Diseases"

_pharmaceuticals, 2021, doi:10.3390/ph14030227_

Round 1
Reviewer 1 Report
This manuscript presents the screening of 600 natural products obtained from temperate plant species against several species of important pathogenic protozoa. Several compounds were shown to have interesting activity and this study highlights the potential of temperate plant species for drug development. The study is well presented and in my opinion is suitable for publication in Pharmaceuticals.
Author Response
Point 1: This manuscript presents the screening of 600 natural products obtained from temperate plant species against several species of important pathogenic protozoa. Several compounds were shown to have interesting activity and this study highlights the potential of temperate plant species for drug development. The study is well presented and in my opinion is suitable for publication in Pharmaceuticals.
Response 1: On behalf of all the authors we would like to thank this reviewer for their very positive comments regarding both the presentation and novelty of this study.
Reviewer 2 Report
Searching the novel chemical compounds as candidate for drugs with anti‐parasitic activities with using PhytoQuest Phytopure library is absolutely justified.
These following concerns need to be addressed:
1) What concentration range of amphotericin B, miltefosine and pentamidine was tested?
2) Please, explain why amphotericin B, miltefosine and pentamidine were used to the study? Please discuss results from previous studies concerning these medicinal substances.
3) Lanosterone‐like sterol derived from the Noble Fir Abies procera (700022) and other the most active compounds can be considered as a source of new anti‐parasitic drugs, however the Authors did not mentioned about state of arts in this aspect. Are the authors' studies first in this research aspect?
4) Why promastigote parasites were cultured with 700022 over 28 weeks? Do the authors have results from earlier time intervals?
5) Was there any time dependent activity?
6) The authors could mention which methods are currently used to treat diseases caused by Plasmodium falciparum, Leishmania mexicana, Trypanosoma evansi and T. brucei.
7) Is the PhytoQuest Phytopure library commercially available ? Does this library have a confirmed chemical structure of purified compounds, i.e. by NMR?
Author Response
Point 1: What concentration range of amphotericin B, miltefosine and pentamidine was tested?
Response 1: This information has been added to the legend of Figure 6 (lines 338 – 341).
Point 2: Please, explain why amphotericin B, miltefosine and pentamidine were used to the study? Please discuss results from previous studies concerning these medicinal substances.
Response 2: Amphotericin B, miltefosine and pentamidine were chosen as they represent current therapies for leishmaniasis, thus representing important benchmarks for this study. This information was previously included in the manuscript at line 342-343. However, to reinforce the importance of these compounds, they are also now described in the introduction of the revised manuscript (lines 82-90).
Point 3: Lanosterone‐like sterol derived from the Noble Fir Abies procera (700022) and other the most active compounds can be considered as a source of new anti‐parasitic drugs, however the Authors did not mentioned about state of arts in this aspect. Are the authors’ studies first in this research aspect?
Response 3: As well as our own study, we cite two other studies where lanosterone‐like sterols have been shown to have antiparasitic activities (lines 192-203). Interestingly, the study in Schistosoma mansoni and Fasciola hepatica focussed on lanosterone‐like sterol derived from the Abies procera as their study led from a screen of a subset of the Phytopure library. Whilst our study is first in many aspects, this is perhaps not so much the case here.
Point 4: Why promastigote parasites were cultured with 700022 over 28 weeks? Do the authors have results from earlier time intervals?
Response 4: The approach adopted followed the approach described by Seifert, K.; Matu, S.; Pérez-Victoria, F.J.; Castanys, S.; Gamarro, F.; Croft, S.L. Characterisation of Leishmania donovani promastigotes resistant to hexadecylphosphocholine (miltefosine). Int. J. Antimicrob. Agents 2003, 22, 380–387, doi:10.1016/S0924-8579(03)00125-0, and cited by us in the paper. The 28 weeks was the time taken to see the clearly defined increase in resistance
(some 8-fold in promastigotes). Yes we have results from earlier timepoints. The main manuscript describes the EC50 of 700022 at 8 weeks in Figure 6C. Further data on the evolution of the change in EC50 for 700022 is shown over time in Figure S3.
Point 5: Was there any time dependent activity?
Response 5: The majority of the study used standard fixed time point assays so we cannot explicitly comment on this. The exception is the BRRoK assay used for the bisobolane sesquiterpene glycosides in Plasmodium falciparum. These experiments use fixed EC50-fold concentrations and are performed over a range of time points. The changes in the loss of bioluminescent signal demonstrate both time- and concentration dependent effects of these compounds against Plasmodium falciparum – general observations first reported in references 52 and 53 cited in the manuscript
Point 6: The authors could mention which methods are currently used to treat diseases caused by Plasmodium falciparum, Leishmania mexicana, Trypanosoma evansi and T. brucei.
Response 6: We have checked and revised the manuscript to ensure this information is provided. Specifically: The use of artemisinin combination therapies for P. falciparum is included in the original manuscript (line 117). For L. mexicana, additional information has been added at lines 84-90 in response to comment 2 above. For T. brucei – we have added additional information and a
new reference (23) at lines 97-99 on melarsoprol treatment for T. b. rhodesiense. For T. evansi, we have included information on treatment using diminazene aceturate (lines 111- 113).
Point 7: Is the PhytoQuest Phytopure library commercially available ? Does this library have a confirmed chemical structure of purified compounds, i.e. by NMR?
Response 7: The PhytoQuest Phytopure library is commercially available, with the author list providing the contact names and addresses for colleagues at PhytoQuest. The chemical structures of compounds in this library have been confirmed by NMR and mass Spectrometry. This information has now been added to the Materials and Methods section (lines 400-402)
Reviewer 3 Report
The article written by Hamza Hameed et al. describe plant natural products as a novel resource for activity against tropical parasitic diseases.
In my opinion, the article is very well written, the research carried out is exhaustive, but I do not understand why the authors examine the toxicity and selectivity of the selected compounds in relation to cancer cell line, e.g. HepG2 (hepatocellular carcinoma), and not in relation to normal cell lines, e.g. HSF or CCD-18Co. The study of compounds on cancer cell lines shows that the compounds show anti-cancer properties, and this was probably not the purpose of the research.
For example line 272, the sentence "Unfortunately, all three compounds had an intrinsic toxicity against HepG2 (CC50 of approximately 2 μM, Table S3) and were not developed any further" is rather misleading.
In my opinion, the authors have shown that these compounds have very good anticancer properties, but they have not shown whether they are toxic to healthy human cells.
Therefore, authors should investigate their toxicity against normal cell lines, which is normal practice when testing new drugs.
Author Response
Response to Reviewer 3 Comments
Point 1: In my opinion, the article is very well written, the research carried out is exhaustive, but I do not understand why the authors examine the toxicity and selectivity of the selected compounds in relation to cancer cell line, e.g. HepG2 (hepatocellular carcinoma), and not in relation to normal cell lines, e.g. HSF or CCD-18Co. The study of compounds on cancer cell lines shows that the compounds show anti-cancer properties, and this was probably not the purpose of the research. For example line 272, the sentence "Unfortunately, all three compounds had an intrinsic toxicity against HepG2 (CC50 of approximately 2 μM, Table S3) and were not developed any further" is rather misleading. In my opinion, the authors have shown that these compounds have very good anticancer properties, but they have not shown whether they are toxic to healthy human cells. Therefore, authors should investigate their toxicity against normal cell lines, which is normal practice when testing new drugs.
Response 1: Thank you for your comments on the manuscript. We agree that using cancer cell lines to define cytotoxicity does not completely reflect the selectivity of these compounds. However, like many others in the field, we have used the cell lines HepG2 and THP1 to obtain preliminary cytotoxicity screening data in order to prioritise those compounds to take forward for further analysis. There are many examples of this practice using one or more of HepG2, THP1, RAW264.7, HeLa and other cancer cell lines. We include a small number of publications here that reflect this to be a contemporary aspect of parasitic screens, including an example of this approach published recently in
Pharmaceuticals:
Peña, I.; Pilar Manzano, M.; Cantizani, J.; Kessler, A.; Alonso-Padilla, J.; Bardera, A.I.; Alvarez, E.; Colmenarejo, G.; Cotillo, I.; Roquero, I.; et al. New compound sets identified from high throughput phenotypic screening against three kinetoplastid parasites: An open resource. Sci. Rep. 2015, 5, doi:10.1038/srep08771.
Roquero, I.; Cantizani, J.; Cotillo, I.; Manzano, M.P.; Kessler, A.; Martín, J.J.; McNamara, C.W. Novel chemical starting points for drug discovery in leishmaniasis and Chagas disease. Int. J. Parasitol. Drugs Drug Resist. 2019, 10, 58–68, doi:10.1016/j.ijpddr.2019.05.002.
Jie Xin, T.; Rajesh Chandramohanadas, K.S.-W.T. High-Content Screening of the Medicines for Malaria Venture Pathogen Box for Plasmodium falciparum Digestive Vacuole-. Antimicrob. Agents Chemother. 2018, 62, 1–17.
Singh, S.; El-sakkary, N.; Skinner, D.E.; Sharma, P.P.; Ottilie, S.; Antonova-koch, Y.;
Kumar, P.; Winzeler, E.; Poonam; Caffrey, C.R.; et al. Synthesis and bioactivity of
phthalimide analogs as potential drugs to treat schistosomiasis, a neglected disease of poverty. Pharmaceuticals 2020, 13, doi:10.3390/ph13020025.
We agree that further cytotoxicity screens should be completed if this research is taken further and we have modified our summary in the revised manuscript to reflect this (lines 372-373). In addition, we have modified the text of the manuscript to make it clear that these are preliminary selectivity data, reflecting that more data would be needed to be unequivocal about cytotoxicity. These changes have been made at lines 208, 243, 285, 297, 302, 346, 391.
Within the time frame, and due to the current national lockdown in England, we have been unable to undertake any further cytotoxicity assays with non-cancer cell lines. However, we hope the changes in the text are sufficient to correct this issue.
Reviewer 4 Report
The authors have presented an interesting and novel research, however, it would have been great if the authors could screen these compounds against at least three human cell lines and present the data along side with the EC50s of the parasites. Could the toxicity be caused by the DMSO, the authors have not stated the concentration of DMSO used. High percent of DMSO could cause cytopathetic effect on both parasites and host cells. It will have been good to check the compounds on normal host cells like HFF and other endothelial cells.
Author Response
Response to Reviewer 4 Comments
Point 1: The authors have presented an interesting and novel research, however, it would have been great if the authors could screen these compounds against at least three human cell lines and present the data along side with the EC50s of the parasites. It will have been good to check the compounds on normal host cells like HFF and other endothelial cells.
Response 1: We agree that further cytotoxicity screens should be completed if this research is taken further and we have modified our summary in the revised manuscript to reflect this (lines 372-373). In addition, we have modified the text of the manuscript to make it clear that these are preliminary selectivity data, reflecting that more data would be needed to be unequivocal about cytotoxicity. These changes have been made at lines 208, 243, 285, 297, 302, 346, 391.
That said, the use of one or two cell lines in preliminary cytotoxic screening is not an uncommon practice and we include here a small number of contemporary/important papers in the field that reflect this approach in parasitic screens. This includes an example of this approach published recently in Pharmaceuticals:
Peña, I.; Pilar Manzano, M.; Cantizani, J.; Kessler, A.; Alonso-Padilla, J.; Bardera, A.I.; Alvarez, E.; Colmenarejo, G.; Cotillo, I.; Roquero, I.; et al. New compound sets identified from high throughput phenotypic screening against three kinetoplastid parasites: An open resource. Sci. Rep. 2015, 5, doi:10.1038/srep08771.
Roquero, I.; Cantizani, J.; Cotillo, I.; Manzano, M.P.; Kessler, A.; Martín, J.J.; McNamara, C.W. Novel chemical starting points for drug discovery in leishmaniasis and Chagas disease. Int. J. Parasitol. Drugs Drug Resist. 2019, 10, 58–68, doi:10.1016/j.ijpddr.2019.05.002.
Jie Xin, T.; Rajesh Chandramohanadas, K.S.-W.T. High-Content Screening of the Medicines for Malaria Venture Pathogen Box for Plasmodium falciparum Digestive Vacuole-. Antimicrob. Agents Chemother. 2018, 62, 1–17.
Singh, S.; El-sakkary, N.; Skinner, D.E.; Sharma, P.P.; Ottilie, S.; Antonova-koch, Y.;
Kumar, P.; Winzeler, E.; Poonam; Caffrey, C.R.; et al. Synthesis and bioactivity of
phthalimide analogs as potential drugs to treat schistosomiasis, a neglected disease of poverty. Pharmaceuticals 2020, 13, 25.
Point 2: Could the toxicity be caused by the DMSO, the authors have not stated the concentration of DMSO used. High percent of DMSO could cause cytopathetic effect on both parasites and host cells.
Response 2: For the initial screen, the DMSO concentration was 1% (v/v), which could show some limited restriction in in vitro parasite culture growth. This effect was controlled using parasites grown in 1% (v/v) DMSO to define 100% growth and all other data were presented relative to this. Whilst not shown here, this effect is rarely >5-10% reduction in parasite growth compared to an untreated control based on several years’ experience in doing these assays. All initial hits were then assayed using serially diluted compound, where EC50 would be determined with solvent concentrations <0.1% of DMSO. The manuscript has been modified at line 413 to reflect that 1% (v/v) DMSO was the concentration used for the initial screens.
Round 2
Reviewer 3 Report
I recommend acceptance of the work in its present form.